# R2-ROUTER: A New Paradigm for LLM Routing with Reasoning

**Jiaqi Xue** [1]   **Qian Lou** [1]   **Jiarong Xing** [2]   **Heng Huang** [3]

## Abstract

As LLMs proliferate with diverse capabilities and costs, LLM routing has emerged by learning to predict each LLM's quality and cost for a given query, then selecting the one with high quality and low cost. However, existing routers implicitly assume a single fixed quality and cost per LLM for each query, ignoring that the same LLM's quality varies with its output length. This causes routers to exclude powerful LLMs when their estimated cost exceeds the budget, missing the opportunity that these LLMs could still deliver high quality at reduced cost with shorter outputs. To address this, we introduce R2-ROUTER, which treats output length budget as a controllable variable and jointly selects the best LLM and length budget, enforcing the budget via length-constrained instructions. This enables R2-ROUTER to discover that a powerful LLM with constrained output can outperform a weaker LLM at comparable cost—efficient configurations invisible to prior methods. Together with the router framework, we construct R2-BENCH, the first routing dataset capturing LLM behavior across diverse output length budgets. Experiments show that R2-ROUTER achieves state-of-the-art performance at $4-5\times$ lower cost compared with existing routers. This work opens a new direction: *routing as reasoning*, where routers evolve from reactive selectors to deliberate reasoners that explore which LLM to use and at what cost budget. The code is publicly available at https://github.com/UCF-ML-Research/R2-Router.

## 1. Introduction

Large language models (LLMs) have shown impressive effectiveness across diverse domains, such as research, in-

---
[1]University of Central Florida [2]Rice University [3]University of Maryland College Park. Correspondence to: Qian Lou <qian.lou@ucf.edu>.

*Proceedings of the 43rd International Conference on Machine Learning*, Seoul, South Korea. PMLR 306, 2026. Copyright 2026 by the author(s).

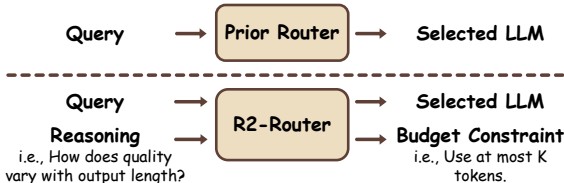

*Figure 1.* Prior routers take a query and select an LLM based on estimated quality and cost. R2-ROUTER additionally reasons about how quality varies with output length, selecting both the best LLM and an appropriate budget constraint.

dustry, and daily applications. This success has catalyzed the emergence of numerous LLMs with diverse architectures, scales, and training objectives, leading to substantial variation in their areas of specialization, response quality, and inference cost. Generally, larger models tend to provide higher quality but come with higher costs, while smaller ones are more affordable but less capable. This diversity creates a new systems-level challenge: deciding, for each query, which model to use to achieve the best balance between quality and cost.

LLM routing has emerged as a promising solution to this challenge, dynamically assigning each query to the most suitable LLM. However, existing routers estimate quality and cost separately, implicitly assuming that each LLM has a single fixed quality–cost profile for each input query. As illustrated in Figure 2 (a), this can lead to suboptimal decisions. For example, powerful LLMs (e.g., Qwen3-235B) may be excluded simply because their estimated cost exceeds the budget. This overlooks intra-model variability: **the same LLM's quality varies with its output length.** By constraining the output length, a powerful LLM can still deliver satisfactory quality at reduced cost. Existing routers are blind to such opportunities because they never reason about: *"How does quality vary with output length?"*

To address this limitation, we propose R2-ROUTER, which introduces a **reasoning**[1] step into routing. As illustrated in Figure 1, R2-ROUTER reasons about the quality each LLM can achieve under different output lengths, then selects the best LLM along with a budget constraint enforced via length-constrained instructions (Lee et al., 2025) (e.g., "use

---
[1]We use "reasoning" by analogy to LLMs like Gemini that dynamically decide thinking depth: our router similarly decides the output length for each LLM to achieve the best balance.

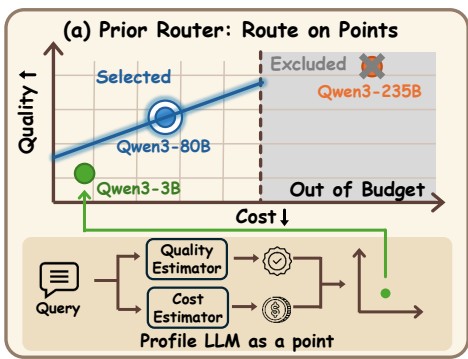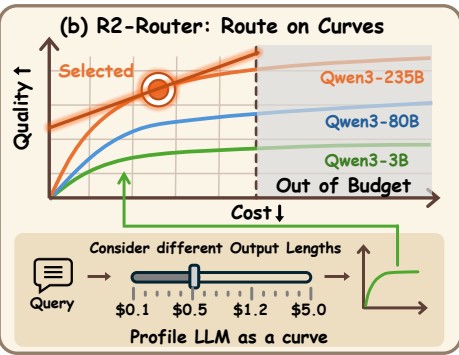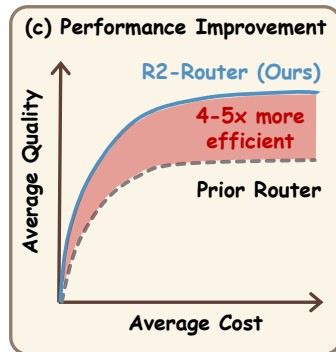

*Figure 2.* (a) Prior Router: Route on Points. Existing routers profile each LLM as a static point, excluding powerful LLMs like Qwen3-235B when their estimated cost exceeds the budget. (b) R2-ROUTER: Route on Curves. By reasoning about cost-dependent quality, R2-ROUTER transforms each LLM from a point into a curve, discovering that powerful LLMs can deliver superior quality at reduced cost. (c) R2-ROUTER achieves comparable quality at $4 - 5\times$ lower cost than prior routers.

at most K tokens"). This reasoning process transforms each LLM from a single quality-cost point into a quality-cost curve, modeling how quality changes as output length varies, as illustrated in Figure 2 (b). Instead of choosing only among LLMs, R2-ROUTER searches over all combinations of LLMs and output lengths. We refer to this as evolving from **routing on points** to **routing on curves**. Through this expanded search space, R2-ROUTER discovers that Qwen3-235B, previously excluded for high cost, can deliver superior quality at reduced cost.

We also introduce R2-BENCH, the first LLM routing dataset that captures LLM behavior across diverse output lengths for the same queries. Unlike existing routing benchmarks that record only a single response per LLM per query, R2-BENCH systematically varies the output token budget, capturing how each LLM's quality evolves with cost. This enables the training and evaluation of routers that can reason about how quality varies with output length.

Extensive experiments demonstrate the effectiveness of our proposed method and dataset from multiple perspectives. First, by capturing quality-cost curves rather than single points, R2-BENCH raises the oracle upper bound by $15\%$ in AUDC, revealing the full potential of routing that prior datasets cannot capture. Second, R2-ROUTER achieves state-of-the-art performance, reaching comparable quality at $4 - 5\times$ lower cost compared with existing methods. Notably, R2-ROUTER is lightweight and data-efficient: by interpolating between only 6 anchor budgets, it approximates continuous quality-cost curves and achieves near-optimal performance with minimal overhead (30 minutes on a single GPU). Finally, R2-ROUTER's reasoning capability is general and can be integrated with various existing routers as a plug-in module. As a case study, we integrate R2-ROUTER with UniRouter (Jitkrittum et al., 2025), a router designed for dynamic LLM pools. When five new models are added to the pool, the integrated router maintains strong adapt-

ability while outperforming the original UniRouter: AUDC improves by $5\%$ and cost drops by $80\%$ to reach comparable quality. Beyond our own benchmarks, R2-ROUTER ranks first on the public RouterArena leaderboard (Lu et al., 2025) at the time of acceptance, confirming its effectiveness against a broad range of competing routers.

**Conflict of Interest Disclosure.** The authors declare no conflicts of interest.

## 2. Related Work

### 2.1. LLM Efficient Reasoning

Recent work has explored how scaling inference-time compute affects LLM reasoning and cost. While longer reasoning generally improves quality, it also increases inference cost. Empirical studies reveal that response quality grows with output length but saturates beyond a threshold (Lee et al., 2025). This observation suggests that by moderately shortening responses through instructional prompts such as "be concise" (Nayab et al., 2024; Jin et al., 2024) or "use at most 100 words" (Han et al., 2024), LLMs can often maintain comparable quality while substantially reducing inference cost.

### 2.2. LLM Routing

Recently, LLM routing has become a fundamental component of modern AI infrastructure. In the open-source ecosystem, platforms such as HuggingFace and vLLM provide routers (vLLM Semantic Router Team, 2025; Hugging Face Router, 2026) that automatically select the best model for specific input. In commercial deployments, OpenAI's GPT-5 (OpenAI, 2025) integrates an internal router to balance capability and pricing tiers, and Azure AI Foundry (Microsoft, 2025) offers a model router as a paid service. Meanwhile, numerous commercial routing platforms such as Open-

Router (2025), NotDiamond (2025), and RequestyAI (2025) continue to emerge, further demonstrating the growing industrial potential of routing in AI infrastructure.

Earlier approaches like FrugalGPT (Chen et al., 2024a) and AutoMix (Aggarwal et al., 2023) cascade queries through models ordered by cost, obtaining responses until one is deemed sufficient. However, this sequential design incurs multiple model calls and introduces latency. In contrast, predictive routing employs data-driven models, including neural networks (Chen et al., 2024b; Šakota et al., 2024; Zhang et al., 2025; 2026; Wu et al., 2026), $k$-nearest neighbors (Somerstep et al., 2025; Hu et al., 2024; Stripelis et al., 2024; Wu & Silwal), and graph neural networks (Feng et al., 2024), to select the optimal LLM by estimating its expected quality and cost. This cost estimation has evolved from relying on static attributes like parameter counts or fixed prices (Song et al., 2025; Shirkavand et al., 2025) to explicitly predicting output lengths (Nguyen et al., 2024; Somerstep et al., 2025; Wu & Silwal) for finer-grained estimation.

Several recent works explore routing beyond model selection. Semantic Router (Wang et al., 2025) and Think When Needed (Guo et al., 2026) select inference modes within a single LLM rather than routing across multiple LLMs. R2-Reasoner (Shao et al., 2025) decomposes a query into multiple reasoning steps and routes each to a different LLM. Route-To-Reason (Pan et al., 2025) extends routing to (LLM, reasoning strategy) pairs, but predicts cost as an outcome of strategy selection without explicit cost control. BEST-Route (Ding et al., 2025) improves the cost–quality trade-off by selecting both the LLM and the number of responses to sample, raising quality at the expense of additional inference calls. HW-Router (Kabir et al., 2026) proposed to considering the live hardware metrics to estimate the cost more accurately. Despite these advancements, all existing frameworks treat each LLM or configuration as a fixed quality-cost point. R2-ROUTER differs fundamentally: by treating output length as a controllable input via length-constrained instructions, we model each LLM as a quality-cost curve rather than a single point. This enables searching over continuous configurations to discover that powerful LLMs with constrained budgets can outperform weaker models—efficient configurations invisible to prior methods.

### 2.3. LLM Routing Dataset

RouterBench (Hu et al., 2024) introduces a large-scale dataset containing over 405k inference outcomes from 11 LLMs. RouterEval (Huang et al., 2025) collects performance results from 8,500 LLMs across 12 widely used benchmarks. SPROUT (Somerstep et al., 2025) aggregates responses from prior datasets under a unified cost annotation framework. Other benchmarks have also contributed

to this line of work by using different data collection methods (Feng et al., 2025; Kassem et al., 2025; Mei et al., 2025; Lu et al., 2025). However, all existing benchmarks ignore how the same LLM behaves under different output lengths. Our R2-BENCH fills this gap by capturing LLM behavior across diverse output lengths, enabling routers to reason about how quality varies with cost.

## 3. Dataset Construction: R2-BENCH

Existing routing benchmarks record only a single response for each LLM on each query, making it impossible to train or evaluate routers that reason about cost-dependent quality. To address this, we construct R2-BENCH, which records multiple responses under different output length budgets for each LLM on each query. We use R2-BENCH for both training and evaluation with a standard train/test split. The statistics of the dataset are detailed in Section 5.1.

### 3.1. Dataset Construction Process

Figure 4 (upper) demonstrates the pipeline of dataset construction. R2-BENCH integrates queries from 6 widely-used benchmarks, including GPQA (Rein et al., 2024), MuSR (Sprague et al., 2024), MMLU-Pro (Wang et al., 2024), MATH (Hendrycks et al., 2021), OpenHermes (Teknium, 2023), and RAGBench (Friel et al., 2024), covering diverse tasks such as reasoning, knowledge understanding, mathematics, and retrieval-augmented generation. We use a fixed pool of 15 open-source LLMs, spanning a range of model sizes (0.6B to 235B), including both general-purpose and domain-specific models (e.g., math-specialized). See Appendix B.2 for the full list; costs follow OpenRouter's official pricing. For each (query, LLM) pair, we collect responses under a set of predefined output length budgets, representing how each model's quality varies with cost. The output lengths are controlled by prompt-based constraints proposed in (Lee et al., 2025; Jin et al., 2024; Nayab et al., 2024; Han et al., 2024), which explicitly instruct the LLM to "use at most $k$ tokens." We empirically validate the effectiveness of this instruction in Appendix A.

To select a reliable LLM judge, we followed the LLM-as-a-judge validation protocol (Zheng et al., 2023) to randomly sample 500 responses and collected human annotations from 30 expert annotators. We then evaluated 4 candidate LLM judges (GLM-4.5-Air, DeepSeek-V3.1, Qwen3-80B-Instruct, and Llama-3.1-70B-Instruct) by computing their Pearson correlation with human judgments. Among them, Qwen3-80B-Instruct achieved the highest correlation ($\rho = 0.82$) and was selected as the final judge for R2-BENCH. This LLM-as-a-judge approach is crucial for evaluating open-ended instruction queries, as traditional automatic evaluation methods like exact match fail to capture response quality in such settings.

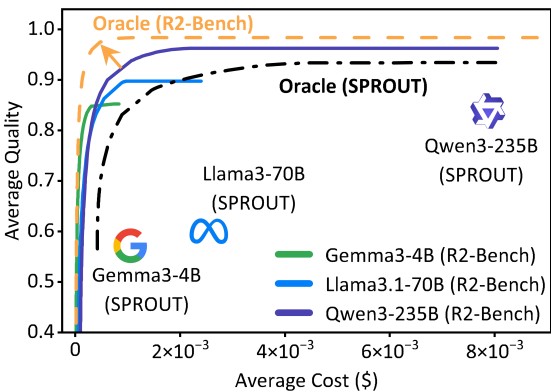

*Figure 3.* Comparison of Oracle performance between R2-BENCH and SPROUT[2] on three representative LLMs. SPROUT profiles each LLM as a single point (icons), while R2-BENCH captures each LLM as a quality-cost curve (solid lines), enabling per-LLM Oracle selection.

### 3.2. Improved Oracle Performance

A key advantage of R2-BENCH over prior single-response datasets is the ability to compute finer-grained Oracle performance. Given a trade-off coefficient $\lambda$, the Oracle selects the response that maximizes $(1 - \lambda)Q - \lambda C$. On prior benchmarks, each LLM has only one response per query, so the Oracle can only select among LLMs. On R2-BENCH, each LLM is represented as a quality-cost curve, enabling the Oracle to select across both LLMs and output lengths.

Figure 3 compares Oracle performance on R2-BENCH and SPROUT[2] with three representative LLMs. R2-BENCH surpasses SPROUT across all metrics: AUDC improves from $0.85$ to $0.98$, QNC decreases from $0.18$ to $0.04$, and Peak Quality increases from $0.90$ to $0.98$. This improvement comes from two sources: (i) per-LLM Oracle, where the Oracle selects the optimal cost for each LLM on the curve; and (ii) the improved per-LLM Oracle raises the overall upper bound, revealing the full potential of reasoning-capable routing. R2-BENCH and R2-ROUTER are thus complementary: R2-BENCH exposes the larger (LLM, budget) optimization space, while R2-ROUTER is the mechanism that searches it. Neither alone yields the gain, as prior routers cannot exploit this space and the space does not exist in prior single-point datasets.

## 4. Routing as Reasoning: R2-ROUTER

### 4.1. Problem Definition

Existing LLM routing is formulated as follows. Let $\mathcal{M} = \{M_1, M_2, ..., M_n\}$ denote a set of LLMs and $\mathcal{X} = \{x_1, x_2, ..., x_m\}$ denote a set of input queries. The goal of

an LLM router is to assign each query $x_i \in \mathcal{X}$ to the most suitable LLM $M_j \in \mathcal{M}$, achieving higher response quality at lower cost. Each candidate LLM is characterized by two quantities: its response quality $Q$ and inference cost $C$. For a given query $x_i$, the router defines a trade-off score that balances quality and cost, i.e., $S = (1-\lambda) \cdot Q - \lambda \cdot C$, where $\lambda$ is a user-defined coefficient that controls cost sensitivity. The router then selects the LLM that maximizes this score.

This formulation assumes each LLM has a fixed $(Q, C)$ pair per query. In the next section, we extend this formulation by treating output length as a controllable variable.

### 4.2. From Reaction to Reasoning

We extend the routing formulation by recognizing that response quality is functionally dependent on output token length, which can be controlled via prompt-based constraints (Lee et al., 2025). We empirically validate this assumption in Appendix A. Let $\mathcal{B} = \{b_1, ..., b_K\}$ be a set of token budgets. An LLM's behavior is therefore not a single static point, but a function of the assigned budget. The routing objective transforms from selecting an LLM to finding the optimal (LLM, token budget) pair $(M^*, b^*)$:

$$(M^*, b^*) = \underset{M \in \mathcal{M}, \, b \in \mathcal{B}}{\arg \max} \, ((1 - \lambda) \cdot Q(x, M, b) - \lambda \cdot C(b)) \tag{1}$$

Here, $C(b)$ denotes the inference cost corresponding to the token budget $b$, calculated as the product of $b$ and the LLM's per-token cost.

This expansion of the search space provides a theoretical guarantee of superior performance compared to traditional methods.

**Definition 4.1** (Reactive Routing). A reactive router predicts a single cost $\hat{c}_i$ (and corresponding quality $\hat{q}_i$) for each LLM $M_i$. Its search space is restricted to a set of fixed operating points:

$$\mathcal{S}_{reactive} = \{(M_i, \hat{c}_i) \mid M_i \in \mathcal{M}\} \tag{2}$$

**Definition 4.2** (Reasoning-based Routing). A reasoning-based router considers a range of feasible token budgets $\mathcal{B}$ for each LLM. Its search space is the Cartesian product of LLMs and budgets:

$$\mathcal{S}_{reasoning} = \{(M_i, b_j) \mid M_i \in \mathcal{M}, b_j \in \mathcal{B}\} \tag{3}$$

**Theorem 4.3** (Optimization Dominance). *Let $S^*(\mathcal{S})$ be the maximum utility achievable within a search space $\mathcal{S}$. Assuming the reactive router's predicted cost $\hat{c}_i$ corresponds to one of the feasible budgets in $\mathcal{B}$, then:*

$$S^*(\mathcal{S}_{reasoning}) \geq S^*(\mathcal{S}_{reactive}) \tag{4}$$

*Proof.* The reactive router locks each LLM to a single predicted cost $\hat{c}_i$. The reasoning-based router searches over all

---

[2]For fairness, we reconstructed SPROUT using the same LLM pool, judge model, queries, and ground truth answers as R2-BENCH.

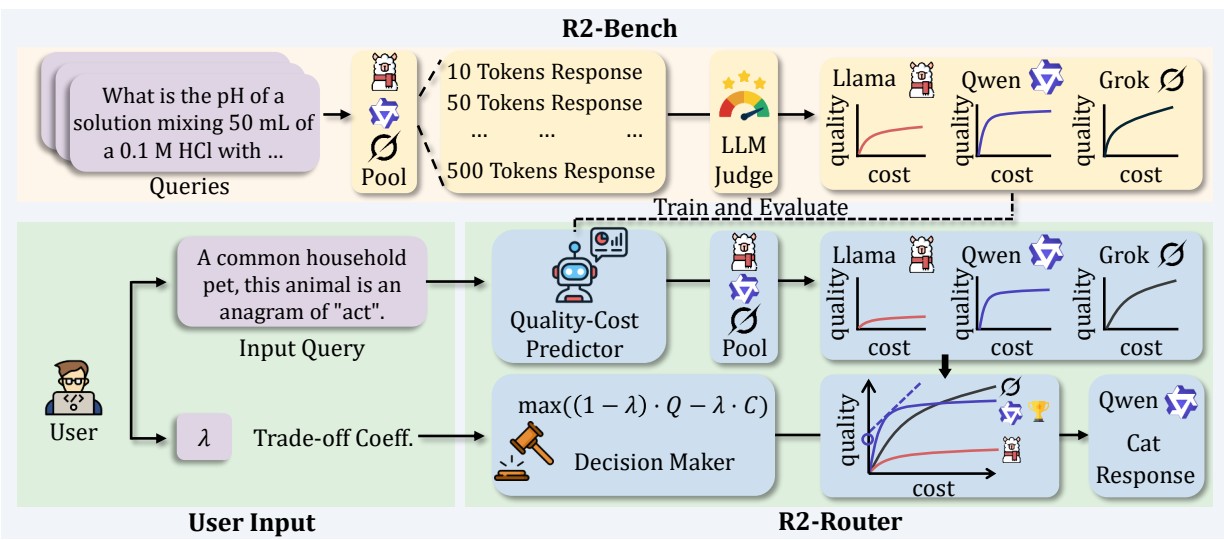

*Figure 4.* An overview of the proposed framework. (Upper) R2-BENCH is built by collecting multiple responses from each LLM under different token budgets (e.g., 10, 50, 500 tokens). An LLM-as-a-judge scores these responses to form quality-cost curves for every (query, LLM) pair. (Lower) R2-ROUTER train a shared encoder with per-LLM, cost-specific quality predictors. At inference time, given a query $x$, a user-defined budget limit $B$, and trade-off coefficient $\lambda$, the router predicts the quality-cost curve for each LLM and selects the optimal (LLM, cost) pair $(M^*, C^*)$ that maximizes $S(x, M, C) = (1 - \lambda) \cdot \hat{Q}(x, M, C) - \lambda \cdot C$.

$b_j \in \mathcal{B}$. Since $\hat{c}_i$ corresponds to one of the feasible budgets, we have $\mathcal{S}_{reactive} \subseteq \mathcal{S}_{reasoning}$. The result follows since the maximum over a superset is at least as large as the maximum over any subset.

Theorem 4.3 highlights that reactive routers constrain the solution space. For example, a powerful $M_{large}$ might be rejected by a reactive router because its predicted cost is too high. However, R2-ROUTER can discover that $M_{large}$ with a constrained budget still outperforms a smaller $M_{small}$ while satisfying the cost constraint. This solution lies in $\mathcal{S}_{reasoning}$ but outside $\mathcal{S}_{reactive}$.

We emphasize that "reasoning" here refers to the *router's* decision process over the (LLM, budget) space, i.e., deliberating about how each LLM's quality would change under different budgets before committing to a choice, rather than to LLM-internal System-2 reasoning such as chain-of-thought. This is the sense in which R2-ROUTER evolves from a reactive selector into a deliberate reasoner.

## 4.3. Methodology of R2-ROUTER

As illustrated in Figure 4 (lower), R2-ROUTER takes a query and a user-specified trade-off coefficient $\hat{\lambda}$ as input. A Quality-Cost Predictor predicts the quality $\hat{Q}(x, M, C)$ that each LLM can achieve at different costs[3], forming a quality-cost curve for each LLM. Then, a Decision Maker

---

[3]Here $C$ denotes the *output cost*, i.e., the output length multiplied by the LLM's per-token price. The input cost (proportional to the input length) is fixed and thus does not affect $\hat{Q}$, but is included as a constant term in the score $S(x, M, C)$.

searches over all LLMs and costs to find the optimal pair that maximizes the trade-off score $S = (1 - \lambda) \cdot Q - \lambda \cdot C$.

### 4.3.1. ARCHITECTURE

R2-ROUTER employs a shared encoder to obtain the query representation $z_x = \text{Enc}(x)$. To approximate the quality-cost curve, R2-ROUTER defines a set of $K$ discrete anchor costs $\mathcal{B} = \{b_1, ..., b_K\}$. For each LLM $M_i$, R2-ROUTER uses a multi-head quality predictor with $K$ independent heads. The $k$-th head $g_{i,k}(\cdot)$ predicts the quality of LLM $M_i$ under cost $b_k$. Each head is a three-layer MLP with ReLU activations and a Sigmoid output to ensure the prediction falls within $[0, 1]$:

$$\hat{Q}(x, M_i, b_k) = \sigma(g_{i,k}(z_x)), \quad \sigma : \mathbb{R} \to [0, 1] \quad (5)$$

This design allows the shared encoder to capture general query semantics, while the independent heads predict LLM behaviors at different costs.

### 4.3.2. TRAINING

The heads are trained using Mean Squared Error (MSE) on R2-BENCH, which provides multiple quality-cost pairs for the same $(x, M)$ under different costs. For a query $x$, LLM $M_i$, and token budget $b_k$, the optimization objective is:

$$\mathcal{L}_{i,k} = \text{MSE}(\hat{Q}(x, M_i, b_k), Q^{\text{true}}(x, M_i, b_k)), \quad (6)$$

$$\theta_{i,k}^* = \arg \min_{\theta_{i,k}} \mathcal{L}_{i,k}, \qquad \forall i, k. \quad (7)$$

Because the predictor is trained on each LLM's observed quality at every budget, R2-ROUTER does not assume that

all queries can be safely compressed: if a query requires a detailed response, the quality at tight budgets is low, and the router accordingly selects a larger budget or a different LLM. The gain therefore comes from exploring what each LLM can achieve at different budgets, not from inducing shorter outputs.

### 4.3.3. ROUTING

**Discrete Routing.** In the basic setting, the routing decision finds the optimal (LLM, token budget) pair $(M^*, b^*)$ from the predefined set $\mathcal{B}$ that maximizes the trade-off score:

$$(M^*, b^*) = \underset{M_i \in \mathcal{M},\, b_k \in \mathcal{B}}{\arg\max} \left( (1 - \lambda) \cdot \hat{Q}(x, M_i, b_k) - \lambda \cdot C(b_k) \right) \tag{8}$$

**Continuous Extension via Interpolation.** Interpolation is an *optional* component: all results in this paper use discrete routing over $K = 16$ anchors (Equation 8), and interpolation only serves to reduce the number of anchors needed when offline data collection is costly. A naive approach to cover the full cost spectrum would require a large number of prediction heads (large $K$), demanding extensive training data. To achieve a large search space with minimal overhead, R2-ROUTER employs a sparse set of anchor costs (small $K$) and bridges them via interpolation. For any token budget $b' \in [b_k, b_{k+1}]$, the quality is estimated as:

$$\hat{Q}(x, M, b') = (1 - \alpha) \cdot \hat{Q}(x, M, b_k) + \alpha \cdot \hat{Q}(x, M, b_{k+1}) \tag{9}$$

where $\alpha = \frac{b' - b_k}{b_{k+1} - b_k}$. This relaxes Equation 8 to search over the continuous spectrum $b' \in [0, B_{user}]$ rather than the discrete set $\mathcal{B}$. After selecting the optimal pair $(M^*, b^*)$, R2-ROUTER invokes LLM $M^*$ and enforces the cost constraint through the prompt instruction "Use at most $b^*$ tokens." following Lee et al. (2025).

### 4.3.4. INTEGRATE WITH EXISTING SCORE COMPUTING

The reasoning capability of R2-ROUTER is orthogonal to existing quality predictors. Whether the quality is estimated through regression (Somerstep et al., 2025), similarity search (Jitkrittum et al., 2025), or probabilistic modeling (Song et al., 2025), R2-ROUTER can extend each method from predicting a single value to predicting a quality-cost curve.

As a case study, we integrate R2-ROUTER with UniRouter (Jitkrittum et al., 2025), which addresses routing with dynamically changing LLM pools where new LLMs may arrive without retraining. UniRouter represents each LLM as a feature vector based on its prediction error on a validation set, enabling generalization to unseen LLMs. R2-ROUTER extends this representation from a single operating point to a quality-cost curve: instead of computing $\Phi(h)$ as the average error at one default cost, we compute $\phi_b(h)$ for

multiple token budgets $b \in \mathcal{B}$, yielding a curve representation for each LLM. This integration improves UniRouter's performance while preserving its ability to handle dynamic LLM pools.

## 5. Experimental Setup

### 5.1. Dataset and Benchmark

We evaluate our method on the proposed R2-BENCH, an extension of the SPROUT benchmark dataset. R2-BENCH contains 30,968 queries spanning 20 categories from 6 diverse benchmarks, covering mathematical reasoning, general knowledge, graduate-level science, and other domains (see Appendix B.1 for details).

For each query, we collect responses from 15 LLMs under 16 different cost levels: {10, 20, 30, 40, 50, 80, 100, 150, 200, 300, 500, 800, 1200, 2000, 4000, default} tokens. The cost is constrained by the instructional prompt "use at most $k$ tokens" and enforced by truncation. The "default" setting allows LLMs to respond without the instructional prompt but with a maximum of 4000 tokens. Each response is annotated with: (i) a quality score (0 to 1) using Qwen3-80B-Instruct as the judge, and (ii) the actual token count consumed during generation.

We use a heterogeneous LLM pool consisting of 15 LLMs, spanning model sizes from 0.6B to 235B parameters. The pool includes both general-purpose models (e.g., GLM-4.6, LLaMA-3.1-70B-Instruct, Qwen3-235B-A22B-Instruct) and domain-specific models (e.g., Qwen2.5-Math-7B-Instruct). The per-token costs follow the price list on OpenRouter; full details can be found in Appendix B.2.

### 5.2. Baseline Methods

We compare R2-ROUTER with top-performing routing methods from the RouterArena benchmark (Lu et al., 2025). All baselines are *reactive* routers that predict fixed quality and cost for each LLM without reasoning about cost-dependent quality.

**MIRT-IRT and NIRT-IRT** (Song et al., 2025): Based on Multidimensional Item Response Theory and Neural Cognitive Diagnosis, respectively. These achieve Top-1 and Top-6 on RouterArena. They predict quality $\hat{Q}_i(x)$ but treat cost as a fixed per-LLM constant $C_i$, selecting the LLM that maximizes $(1 - \lambda) \cdot \hat{Q}_i(x) - \lambda \cdot C_i$.

**CARROT-KNN (-K) and CARROT-Linear (-L)** (Somerstep et al., 2025): Top-3 on RouterArena. CARROT uses separate predictors for quality $\hat{Q}_i(x)$ and output token count $\hat{T}_i(x)$, selecting LLMs by maximizing $(1 - \lambda) \cdot \hat{Q}_i(x) - \lambda \cdot C_i \cdot \hat{T}_i(x)$, where $C_i$ is the per-token price. We evaluate two variants using $k$-nearest neighbor ($k = 5$) and linear regression.

**UniRouter** (Jitkrittum et al., 2025): Designed for dynamic LLM pools where new LLMs may arrive without retraining. UniRouter represents each LLM as a feature vector based on its prediction error on a validation set, enabling generalization to unseen LLMs. We also evaluate UNI-R2ROUTER, which integrates R2-ROUTER's reasoning capability with UniRouter (Section 4.3.4).

### 5.3. Implementation Details

Queries are encoded into 1024-dimensional embeddings using Qwen3-Embedding-0.6B. These embeddings serve as input features for both baselines and R2-ROUTER. For R2-ROUTER, we train a separate three-layer MLP for each cost level, with hidden dimensions $[256, 128, 64]$. Each MLP predicts the quality score at its designated cost. All predictors are optimized using MSE loss with Adam optimizer (learning rate $1 \times 10^{-4}$) for 100 epochs.

### 5.4. Computational Resources

The construction of R2-BENCH requires collecting responses from multiple LLMs under different token budgets. While we used 8 NVIDIA B200 GPUs for efficient data collection, the dataset can alternatively be constructed via API calls (e.g., through OpenRouter). In contrast, training R2-ROUTER itself is lightweight since it only involves MLPs without any large language models. Using a single NVIDIA RTX 3090, training routers for 15 LLMs takes approximately 30 minutes. At inference time, the routing overhead is negligible: routing a single query takes less than 400 ms on average, accounting for less than 1% of the total LLM generation time. R2-ROUTER also scales cheaply to a growing LLM pool: adding a new model requires only a lightweight validation-set profiling (e.g., $< \$50$ in API cost plus ~30 min of training on a single RTX 3090 to add Gemini 3 Pro Preview), and re-estimation is needed only when a model is updated or the query distribution shifts substantially.

### 5.5. Evaluation Metrics

We evaluate each routing strategy using a deferral curve (Jitkrittum et al., 2025), which plots the average response quality against the total inference cost (input plus output tokens). While R2-ROUTER optimizes over output cost, since it is the component that varies with the budget, all reported costs are total costs, so the $4$–$5\times$ reduction is measured on total rather than output-only cost. Sweeping the routing penalty parameter $\lambda$ over the interval $[0, 1]$ (Equation 8) traces the deferral curve. Following (Jitkrittum et al., 2025), we employ three evaluation metrics: (i) Area Under the Deferral Curve (AUDC), measuring the overall quality-cost trade-off; (ii) Peak Quality, the maximum achievable quality; and (iii) Query-Normalized Cost (QNC), the min-

imum relative cost required to match the performance of the most accurate LLM in the pool. All experiments are conducted over 5 independent runs with different random seeds, and we report the mean and standard deviation of the results.

## 6. Results

### 6.1. Main Results

**R2-ROUTER achieves state-of-the-art Performance.** As shown in Figure 5, R2-ROUTER consistently achieves higher quality at any given cost compared to all baselines. Notably, R2-ROUTER reaches an average quality of 0.8 at a minimal cost of approximately $0.5 \times 10^{-3}$. In contrast, reactive baselines require $4\times$ to $5\times$ that budget to match similar performance. By reasoning about cost-dependent quality, R2-ROUTER identifies efficient operating points (e.g., powerful LLMs with constrained costs) that reactive routers inherently overlook. We further corroborate these gains on the RouterArena benchmark (Lu et al., 2025), where, under exact-match scoring (no LLM judge) and an expanded pool of reasoning-capable LLMs, R2-ROUTER leads all baselines across all 7 categories; it also extends naturally to thinking-token budgets on reasoning LLMs (Appendix C).

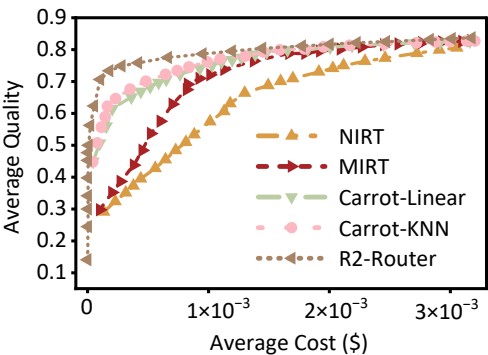

*Figure 5.* Deferral curves comparing routing methods. R2-ROUTER achieves consistently higher quality at lower costs than point-based routing methods.

### 6.2. Generalization to New LLMs

To evaluate generalization, we simulate a dynamic setting transitioning from an initial pool of 6 LLMs to an expanded pool with 5 unseen additions. UniRouter (Jitkrittum et al., 2025) generalizes by embedding validation error to profile the capabilities of each LLM. We integrate this mechanism into R2-ROUTER (termed UNI-R2ROUTER) to predict quality-cost curves across 16 token budgets. Figure 6 demonstrates that UNI-R2ROUTER effectively leverages the expanded pool to significantly outperform the point-based UniRouter, achieving a higher AUDC (0.623 vs. 0.590) while reducing QNC by 80%. This validates our proposition in Section 4.3.4 that R2-ROUTER enhances existing frameworks' efficiency while preserving dynamic adaptability.

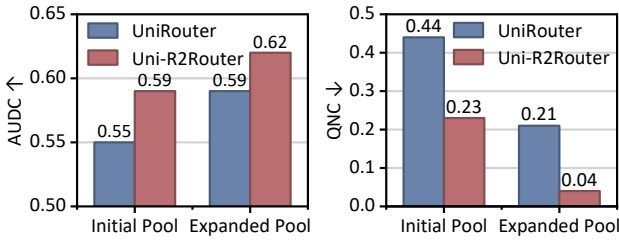

*Figure 6.* Generalization Capabilities. This experiment compares routing on an Initial Pool (GLM-4.6, Llama-3.1-70B, gemma-3-4b, Qwen2.5-Math-1.5B, Qwen3-0.6B, gemma-3-270m) versus an Expanded Pool that adds 5 unseen models (GLM-4.5-Air, Mistral-7B-v0.2, Qwen2.5-Math-7B, Llama-3.2-3B, gemma-3-1b).

### 6.3. Generalization to Out-of-Distribution Queries

To evaluate OOD generalization, we split MMLU-Pro into STEM (training) and non-STEM (testing) disciplines. As shown in Table 1, R2-ROUTER outperforms all baselines, achieving an AUDC of 0.71 compared to 0.67 for the next-best method CARROT-L. This demonstrates R2-ROUTER's cost-efficiency and robustness even under significant distributional shifts.

*Table 1.* Performance comparison on OOD queries.

|  | AUDC ↑ | QNC ↓ | Peak Acc. ↑ |
|---|---|---|---|
| MIRT | $0.63_{\pm0.05}$ | $0.72_{\pm0.08}$ | $0.81_{\pm0.00}$ |
| CARROT-L | $0.67_{\pm0.04}$ | $0.56_{\pm0.06}$ | $0.82_{\pm0.00}$ |
| R2-ROUTER | $0.71_{\pm0.05}$ | $0.26_{\pm0.09}$ | $0.83_{\pm0.00}$ |

### 6.4. Interpolation with Limited Points

As proposed in Section 4.3.3, R2-ROUTER can approximate continuous quality-cost curves via piecewise linear interpolation over a discrete set of anchor points. Figure 7 analyzes the sensitivity of routing performance to the number of trained heads $K$. We observe that R2-ROUTER is highly data-efficient: QNC converges rapidly, reaching near-optimality ($\approx 0.12$) with as few as 6 to 8 anchor points. Crucially, even a coarse approximation with $K = 4$ significantly outperforms point-based baselines like MIRT (QNC=0.43) and CARROT-L (QNC=0.32). This validates that a minimal set of trained heads is sufficient to effectively model the quality-cost landscape.

### 6.5. Ablation Study

To verify the robustness of R2-ROUTER to component choices, we conduct ablation studies on the embedding model and the quality predictor architecture.

**Robustness to Embedding Models.** Our default implementation utilizes `Qwen3-Embedding-0.6B` to extract query features. We investigate the impact of the semantic encoder by substituting it with a smaller alternative,

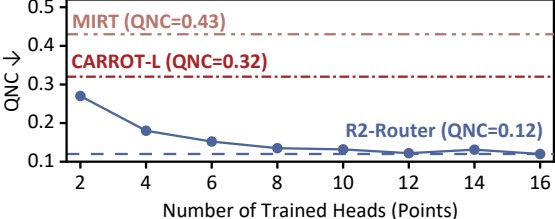

*Figure 7.* Performance comparison of R2-ROUTER with varying numbers of interpolation heads ($K$) against point-based baselines. R2-ROUTER approximates the continuous quality-cost curve via piecewise linear interpolation using the $K$ anchor points.

`MiniLM-L6-v2`. As shown in Table 2, although the absolute metric values fluctuate due to different embedding dimensions and capabilities, R2-ROUTER consistently maintains a substantial lead over point-based baselines. For instance, R2-ROUTER achieves the lowest QNC of 0.32 with MiniLM, demonstrating that our framework generalizes well across different embedding spaces.

*Table 2.* Using `MiniLM-L6-v2` as the embedding model.

|  | AUDC ↑ | QNC ↓ | Peak Acc. ↑ |
|---|---|---|---|
| MIRT | $0.71_{\pm0.05}$ | $0.51_{\pm0.03}$ | $0.79_{\pm0.00}$ |
| CARROT-L | $0.72_{\pm0.02}$ | $0.48_{\pm0.02}$ | $0.79_{\pm0.00}$ |
| R2-ROUTER | $0.76_{\pm0.04}$ | $0.32_{\pm0.04}$ | $0.79_{\pm0.00}$ |

**Impact of Predictor Choice.** We typically employ a 3-layer MLP as the quality predictor. To study the influence of the predictor architecture, we experiment with a gradient-boosting decision tree-based regressor (LGBM). The results in Table 3 indicate that R2-ROUTER is model-agnostic regarding the regression head. Notably, using LGBM further boosts the AUDC to 0.80 while maintaining a low QNC of 0.29, confirming that R2-ROUTER's performance gains stem from the curve-based optimization strategy rather than a specific predictor architecture choice.

*Table 3.* Performance using LGBM as prediction head.

|  | AUDC ↑ | QNC ↓ | Peak Acc. ↑ |
|---|---|---|---|
| MIRT | $0.74_{\pm0.03}$ | $0.78_{\pm0.02}$ | $0.81_{\pm0.00}$ |
| CARROT-L | $0.77_{\pm0.02}$ | $0.66_{\pm0.02}$ | $0.80_{\pm0.00}$ |
| R2-ROUTER | $0.80_{\pm0.03}$ | $0.29_{\pm0.03}$ | $0.81_{\pm0.00}$ |

**Robustness to Judge Selection.** To mitigate potential bias from the LLM judge, we additionally evaluate all methods using `DeepSeek-V3.1` as the judge at test time, while keeping the training labels from `Qwen3-80B-Instruct` unchanged. As shown in Table 4, R2-ROUTER consistently outperforms reactive baselines under this different judge, demonstrating robustness to the choice of evaluation model.

**Impact of Length-Constraint Prompts.** To verify that our

*Table 4.* Performance using `DeepSeek-V3.1` as the LLM judge.

| | AUDC ↑ | QNC ↓ | Peak Acc. ↑ |
|---|---|---|---|
| MIRT | $0.76_{\pm 0.03}$ | $0.55_{\pm 0.02}$ | $0.88_{\pm 0.00}$ |
| CARROT-L | $0.75_{\pm 0.04}$ | $0.52_{\pm 0.03}$ | $0.87_{\pm 0.00}$ |
| R2-ROUTER | $0.80_{\pm 0.04}$ | $0.35_{\pm 0.01}$ | $0.90_{\pm 0.00}$ |

efficiency gains stem from reasoning-based routing rather than merely prompting models to be concise, we apply the length-constraint prompt (e.g., "Be concise") to the input of the reactive baseline. The results in Table 5 show that R2-ROUTER consistently outperforms the prompted baseline. The reason is that while length-constraint prompts affect LLM outputs, they do not influence the selection logic of reactive routers. Reactive routers still rely on static cost estimates, assuming large models are inherently expensive and excluding them from consideration. Consequently, efficient configurations involving powerful LLMs are never selected, whereas R2-ROUTER correctly identifies and leverages these opportunities.

*Table 5.* Comparison against prompt-augmented baselines.

| | AUC ↑ | QNC ↓ | Peak Acc. ↑ |
|---|---|---|---|
| MIRT | $0.77_{\pm 0.03}$ | $0.70_{\pm 0.03}$ | $0.82_{\pm 0.00}$ |
| CARROT | $0.78_{\pm 0.03}$ | $0.68_{\pm 0.02}$ | $0.79_{\pm 0.00}$ |
| R2-ROUTER | $0.83_{\pm 0.02}$ | $0.25_{\pm 0.00}$ | $0.83_{\pm 0.00}$ |

## 7. Conclusion

We identify a key limitation of existing LLM routers: they assume each LLM has only one fixed behavior, overlooking that quality varies with output length. To address this, we propose R2-ROUTER, which reasons about each LLM's quality across different token budgets, transforming routing from selecting among fixed points to searching over quality-cost curves. We also introduce R2-BENCH, the first routing dataset capturing LLM behavior across diverse token budgets. Experiments show that R2-ROUTER achieves state-of-the-art performance at $4 - 5\times$ lower cost, generalizes well to new LLMs and out-of-distribution queries, and can be integrated with existing routers as a plug-in module.

## Acknowledgements

The authors thank the anonymous reviewers for their constructive feedback. This material is based upon work supported in part by the National Science Foundation (NSF) under Grant Nos. CCF-2523407 and CNS-2413232. Jiaqi Xue and Qian Lou are partially supported by these grants. H. Huang was partially supported by NSF IIS-2347592, IIS-2348169, DBI-2405416, CCF-2348306, CNS-2347617, and RISE-2536663.

## Impact Statement

This paper presents work whose goal is to advance the field of Machine Learning, specifically improving the cost-efficiency of LLM inference through intelligent routing. By enabling high-quality responses at reduced computational cost, our work may contribute to making LLM capabilities more accessible and reducing the environmental footprint of AI systems.

Beyond output length, the core idea of treating LLM behavior as controllable rather than fixed may extend to other variables that influence quality and cost, such as system prompts, decoding strategies, or reasoning depth. We hope this work inspires further research on reasoning-based routing that explores the full space of LLM configurations. We do not foresee any specific negative societal consequences that must be highlighted here.

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

## A. Effectiveness of Length-Constrained Instructions

A core assumption of R2-ROUTER is that output length can be controlled via prompt-based instructions. To validate this, we analyze the compliance rate (percentage of responses where actual length $\leq 1.1 \times$ budget) across all 15 LLMs and 5 token budgets. Figure 8 presents the compliance heatmap. We observe two key findings:

*(1) Large models reliably follow length constraints.* Qwen3-235B and DeepSeek-V3 achieve compliance rates above 82% even at the most restrictive budget (10 tokens), and above 97% at moderate budgets ($\geq$100 tokens).

*(2) Small models show lower compliance at tight budgets.* Models below 4B struggle significantly, with compliance dropping to 3%–21% at budget 10.

Crucially, finding (2) does not undermine our method. The key insight of R2-ROUTER is to *unlock the potential of powerful LLMs at reduced cost*—not to make small models cheaper. Small models are already inexpensive, so there is little benefit in constraining their output further. The value of R2-ROUTER lies in discovering that large models with constrained budgets can outperform small models at comparable cost. Finding (1) validates exactly this: powerful LLMs can be effectively constrained to operate at reduced cost while maintaining controllability.

Furthermore, R2-ROUTER naturally learns to avoid unreliable configurations. Since the router is trained on actual responses under length constraints, it implicitly learns each model's compliance behavior. When a model frequently exceeds its budget, the observed cost is higher than intended, and the learned quality-cost curve reflects this reality.

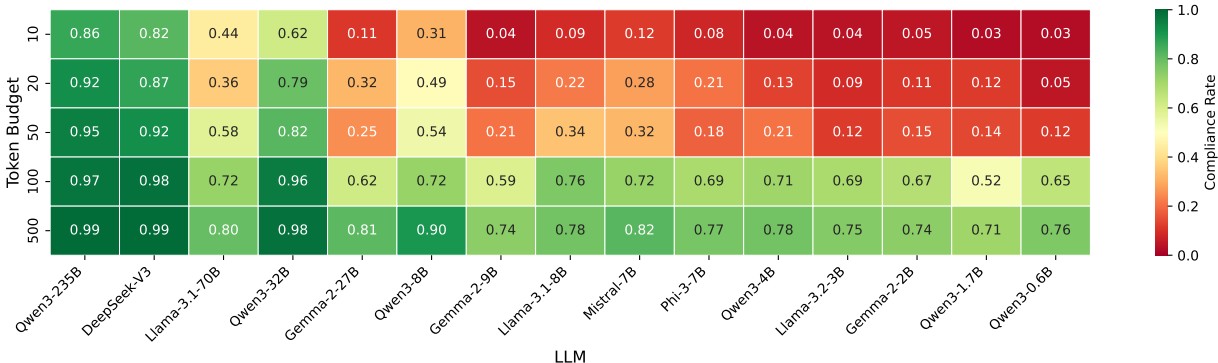

*Figure 8.* Compliance rate of length-constrained instructions across 15 LLMs and 5 token budgets. Large models (left) reliably follow length constraints, validating the core mechanism of R2-ROUTER. Lower compliance of smaller models at tight budgets does not affect our method, as R2-ROUTER's value lies in unlocking powerful LLMs at reduced cost.

## B. Details of Experiments

### B.1. Datasets Composition

R2-BENCH integrates queries from 6 widely-used benchmarks spanning 20 categories:

- MMLU-Pro(Wang et al., 2024) (8,264 queries): Professional-level multiple-choice questions across STEM, humanities, social sciences, and other areas.
- OpenHermes(Teknium, 2023) (13,670 queries): General knowledge and reasoning tasks.
- MATH(Hendrycks et al., 2021) (5,122 queries): Mathematical problem-solving.
- GPQA(Rein et al., 2024) (384 queries): Graduate-level science questions.
- MuSR(Sprague et al., 2024) (100 queries): Multi-step soft reasoning tasks.
- RAGBench(Friel et al., 2024): Retrieval-augmented generation tasks.

### B.2. LLM Pool Details

Table 6 lists the 15 LLMs used in R2-BENCH, along with their per-token costs from OpenRouter.[4] The pool spans model sizes from 0.6B to 235B parameters, including both general-purpose models and domain-specific ones (e.g., Qwen2.5-Math

---

[4] https://openrouter.ai/

for mathematical reasoning). Pricing is subject to change; values shown were retrieved in January 2026.

*Table 6.* LLM pool used in R2-BENCH with per-token pricing.

| LLM | Input ($/1M) | Output ($/1M) |
| --- | --- | --- |
| Qwen3-0.6B | 0.07 | 0.46 |
| Gemma-3-1B | 0.01 | 0.04 |
| Qwen2.5-Math-1.5B-Instruct | 0.01 | 0.02 |
| LLaMA-3.2-3B-Instruct | 0.02 | 0.02 |
| Gemma-3-4B-it | 0.02 | 0.07 |
| Mistral-7B-v0.2 | 0.20 | 0.20 |
| Qwen2.5-Math-7B-Instruct | 0.03 | 0.09 |
| GLM-4.5-Air | 0.35 | 1.55 |
| GLM-4.6 | 0.44 | 1.76 |
| LLaMA-3.1-70B-Instruct | 0.12 | 0.30 |
| Qwen3-235B-A22B-Instruct | 0.18 | 0.54 |

## C. Additional Experiments

This section reports additional experiments conducted during the review period. Unless otherwise stated, all results use discrete routing over the $K = 16$ anchor budgets (Equation 8) without interpolation.

### C.1. Evaluation on RouterArena

To assess R2-ROUTER on a broader set of tasks and to rule out any bias from the LLM judge, we additionally evaluate on RouterArena (Lu et al., 2025), a comprehensive routing benchmark covering $8,400$ queries across 7 diverse categories and 5 evaluation dimensions (accuracy, cost, optimality, robustness, and latency). Unlike R2-BENCH, RouterArena scores responses via exact match against ground-truth answers, so no LLM judge is involved, directly eliminating any intra-family judge bias. We further expand the LLM pool to include reasoning-capable models (e.g., Gemini-2.5-Flash and Claude-3). As shown in Table 7, R2-ROUTER outperforms the strongest baseline (CARROT-L) across all 7 categories, with gains ranging from $+4.1$ to $+8.1$.

*Table 7.* Per-category Arena Scores on RouterArena (Lu et al., 2025). R2-ROUTER leads in every category under exact-match scoring (no LLM judge).

| Method | Knowledge | Math | Domain | Code | Trivia | Translation | NLU |
| --- | --- | --- | --- | --- | --- | --- | --- |
| R2-ROUTER | 78.2 | 76.0 | 71.5 | 69.7 | 66.5 | 65.3 | 64.2 |
| CARROT-L | 73.7 | 67.9 | 65.4 | 62.0 | 61.9 | 60.2 | 60.1 |
| *Gap* | +4.5 | +8.1 | +6.1 | +7.7 | +4.6 | +5.1 | +4.1 |

### C.2. Extension to Reasoning LLMs

R2-ROUTER controls reasoning capacity, not merely answer verbosity. For non-reasoning LLMs, the reasoning process is expressed directly in output tokens, so constraining output tokens constrains reasoning. For reasoning LLMs, reasoning capacity is governed by the thinking-token budget, which several providers already expose as a controllable parameter: Anthropic (`budget_tokens`, 1K–128K), Google Gemini 2.5 (`thinkingBudget`, 128–32K), and Qwen3 (`thinking_budget`, 1–82K). R2-ROUTER's pipeline extends to these models directly: we train it with thinking-token budgets as anchor points on reasoning-capable models (including Gemini-2.5 and Qwen3-235B), learning to predict both quality and cost at each thinking budget and enabling the same curve-based routing as for output tokens. On RouterArena with this extended pool, R2-ROUTER achieves an Arena Score of 71.6, compared to 66.9 (MIRT) and 63.9 (CARROT-L).

### C.3. Robustness to Model-Specific System Prompts

In practice, R2-ROUTER's budget instruction ("use at most $k$ tokens") can be combined with any model-specific system prompt, as nearly all LLM APIs support custom system prompts. The two serve different purposes: model-specific prompts define style and format, while the budget instruction guides output length. Any interaction between them is implicitly captured by the quality predictor, which is trained on actual length-guided outputs under the same prompting setup. To validate this, we conduct an experiment in which each LLM receives its own system prompt (e.g., Qwen3-235B: "Respond in structured format"; Gemini-2.5: "Please be thorough and cite sources"; Llama-3.1-70B: "Respond concisely without step-by-step reasoning"), and all routers are trained and evaluated under this heterogeneous per-LLM prompting setup. R2-ROUTER (AUDC 0.81) still outperforms the best baseline (AUDC 0.76), confirming that the routing advantage holds even when each LLM operates under its own system prompt.

### C.4. Interpolation: PCHIP vs. Piecewise Linear

Interpolation is an *optional* component of R2-ROUTER: all results in the main text are obtained via discrete routing over $K = 16$ anchor budgets without interpolation (Section 4.3.3). Interpolation is studied separately to explore whether the number of required anchors can be reduced, lowering offline data-collection cost while maintaining routing performance. Because the quality–cost relationship can be non-convex, piecewise linear interpolation (PLI) introduces approximation error in such regions; this is a deliberate trade-off between accuracy and data efficiency. We compare PLI against PCHIP (monotone cubic interpolation) at non-anchor budgets using $K = 8$ anchors (Table 8). PCHIP achieves both lower prediction error and better routing QNC. We therefore adopt PCHIP for interpolation. Notably, even PLI already far outperforms all point-based baselines (QNC 0.14 vs. MIRT 0.43, CARROT-L 0.32).

*Table 8.* PLI vs. PCHIP interpolation at non-anchor budgets ($K = 8$ anchors). Lower is better for both metrics.

| Interpolation | Prediction Error | QNC |
|---|---|---|
| Piecewise Linear (PLI) | 0.09 | 0.14 |
| PCHIP (monotone cubic) | **0.05** | **0.12** |

