# OpenReview forum: "R2-Router: A New Paradigm for LLM Routing with Reasoning"
_ICML.cc/2026/Conference — ICML 2026 regular_

### Official Review · Reviewer_jomN · 2026-03-09

**Soundness:** 3
**Presentation:** 2
**Significance:** 2
**Originality:** 3
**Overall Recommendation:** 4
**Confidence:** 3

**Summary:**

This manuscript points out the limitations of existing LLM routing systems that treat models as static "quality-cost" single points, and proposes the R2-ROUTER framework, which incorporates the "output token budget" as a controllable independent variable into the joint optimization space. By employing MLPs to predict the quality at discrete budget anchors and supplementing this with piecewise linear interpolation to construct continuous quality-cost curves, this method dynamically allocates the optimal "model and budget" combination for a given query. Concurrently, the authors constructed R2-BENCH, the first multi-budget routing evaluation benchmark; experiments demonstrate that this "curve routing" paradigm can achieve an approximately 4- to 5-fold cost reduction compared to traditional static routing while maintaining generation quality.

**Compliance With Llm Reviewing Policy:**

Affirmed.

**Key Questions For Authors:**

The core assumption of this manuscript is that constraining the output length (token budget) can reduce costs while maintaining a relatively high quality score. However, from the perspective of real-world user interaction, "shortening the output" is not always a lossless compression process. The adequacy of the output length is highly dependent on the user's underlying intent: if the user seeks rapid fact retrieval, a concise response is optimal; conversely, if the user's goal is to request the large language model to teach specific knowledge in detail to facilitate information integration (e.g., conceptual explanations, step-by-step coding tutorials, or complex logical derivations), forcefully compressing the output length will cause the response to lose its core educational significance and practical value. Does the Quality-Cost Predictor of R2-ROUTER possess an "intent-aware" capability?  When encountering a query that explicitly demands a lengthy and exhaustive explanation, is a forcefully imposed low budget severely penalized within this scoring system?

**Limitations:**

1. Thoroughly supplement the discussion on limitations:
- Omission of input costs: Treating the input cost as a constant will severely dilute the claimed benefit of a 4-5 cost reduction in long-context scenarios (such as RAGBench).
- Lack of task granularity awareness: Forcibly compressing the token budget will cause complex tasks, such as in-depth reasoning and conceptual analysis, to lose their core value.
- Single-judge bias: Relying exclusively on Qwen3-80B-Instruct as the judge is highly susceptible to introducing intra-family bias, thereby limiting the absolute objectivity of the benchmark rankings.

**Strengths And Weaknesses:**

**Strengths：**
- 1. Expansion of the routing search space: Incorporating the token budget as an independent variable into the optimization space transforms the traditional "static single-point routing" into a "dynamic routing based on cost-quality curves," providing heuristic value for system design.
- 2. Effective optimization of inference cost: Empirical results demonstrate that this framework achieves an approximately 4- to 5-fold reduction in cost while maintaining equivalent generation quality, exhibiting strong feasibility for practical deployment.
- 3.  Construction of an evaluation benchmark: The R2-BENCH dataset is proposed, providing a standardized testing benchmark for evaluating LLM routing performance across multiple output length budgets.

**Weaknesses：**
- 1. This manuscript emphasizes "Routing with Reasoning," yet its actual algorithm merely consists of MLP-based cost-quality regression prediction and budget-constrained heuristic search, which lacks the widely recognized logical reasoning processes (e.g., CoT) within the LLM context, thus constituting a conceptual overclaim. It is highly recommended to comprehensively revise the misleading terminology (e.g., substituting it with objective descriptions such as "Budget-Aware/Length-Constrained Routing"); however, should the original nomenclature be retained, the authors must strictly delineate the fundamental distinction between the "heuristic search" employed in this study and the traditional "System-2 reasoning" of LLMs to eliminate conceptual ambiguity in the academic context.
- 2. Risk of Intra-family Bias in the evaluation benchmark: R2-BENCH exclusively employs Qwen3-80B-Instruct as the judge model, while the candidate pool happens to include Qwen3-235B from the same model family; this setup is highly susceptible to triggering the well-documented "self-preference bias," leading to severe systematic deviations in the Ground Truth. It is recommended to introduce a multi-judge consensus mechanism by employing at least two capable models outside the Qwen family (e.g., GPT-4o or Llama-3.1-405B) to rescore the core subset in order to verify the robustness of the rankings.
- 3.  Lack of Theoretical Support and Intent Awareness for linear interpolation: The manuscript's use of piecewise linear interpolation to approximate the quality-cost curve presents critical flaws: on one hand, the process of restricting tokens to increase information density is highly non-linear (quality collapses precipitously when crossing a critical threshold), meaning that linear fitting profoundly violates the true generation logic of LLMs and is highly prone to overestimating the quality at intermediate budgets; on the other hand, the framework erroneously assumes that all queries can be safely compressed, ignoring the task granularity discrepancies where the forced compression of complex reasoning tasks would result in the loss of their core value.  It is recommended to empirically evaluate the true prediction errors of non-linear functions (such as sigmoid or logarithmic models) at non-anchor budgets, and to conduct stratified validation on the test data based on "query granularity requirements" (e.g., detail-oriented vs. summary-oriented) to demonstrate the framework's robustness in tasks requiring detailed explanations.

---

> ### Author Rebuttal · Authors · 2026-03-31
>
> We thank the reviewer for the positive evaluation. We address the detailed suggestions below and revise our manuscript accordingly.
>
> ### Q1: "Routing with Reasoning" is a conceptual overclaim
>
> **"Reasoning" in R2-Router refers to the router's decision process, not to LLM-internal CoT.** We acknowledge that "reasoning" in the LLM context typically denotes System-2 processes such as CoT. We chose "reasoning" to convey a fundamental paradigm shift: prior routers assume each LLM has a single fixed behavior, whereas R2-Router reasons about how the same LLM behaves differently under different budgets — asking "how does this LLM's quality change with budget?" before deciding, exploring configurations that reactive routers never consider. We will add a paragraph in Section 4.2 explicitly stating that R2-Router's "reasoning" is about the router's decision process over the (LLM, budget) space, distinct from LLM-internal System-2 reasoning such as CoT.
>
> ### Q2: Intra-family bias (Qwen judge + candidate)
>
> **R2-Router's advantage holds even without any LLM judge.** To further address this concern, we evaluated on RouterArena[1], which uses exact-match scoring against ground-truth answers (no LLM judge involved), and R2-Router maintains its advantage. Additionally, we addressed this concern in our manuscript: Table 4 reports an ablation using DeepSeek-V3.1 as judge, whose model family is entirely absent from our LLM pool, ensuring zero intra-family bias.
>
> ### Q3: Linear interpolation lacks theoretical support
>
> **We agree that interpolation accuracy is an important consideration; notably, interpolation is an optional component in R2-Router, not a required one.** All improvements in the results are achieved via discrete routing over K=16 anchor budgets (Eq. 8) without interpolation. Interpolation is studied separately in Figure 7 to explore whether it can reduce the number of required anchors, thereby lowering offline data collection costs while maintaining routing performance. We agree that the quality-cost relationship is often non-convex, and piecewise linear interpolation introduces error in such cases — this is a deliberate trade-off between approximation accuracy and data efficiency. Figure 7 shows this trade-off is favorable: K=6 approximates the full 16-anchor performance, and even K=4 outperforms all baselines (QNC 0.15 vs. MIRT 0.43, CARROT-L 0.32). Following the reviewer's suggestion, we further compared PLI against PCHIP (monotone cubic) at non-anchor budgets using K=8 anchors: PCHIP achieves lower prediction error (0.05 vs 0.09) and better routing QNC (0.12 vs 0.14). We will adopt PCHIP in the revision. Notably, even PLI (QNC 0.14) already far outperforms all baselines (MIRT 0.43, CARROT-L 0.32).
>
> ### Q4: Intent-awareness: Does R2-Router assume all queries can be safely compressed?
>
> **R2-Router does not assume all queries can be compressed — and low budgets are penalized for queries that need detailed responses.** The quality predictor is trained on each LLM's observed quality across different budgets for a given query. If a query requires a detailed explanation but an LLM can only produce a short answer within a tight budget, the quality score drops — the router then selects a higher budget or a different LLM. The improvement comes from exploring what each LLM can achieve at different token budgets and selecting the optimal (LLM, budget) configuration — not from inducing shorter outputs. We verified this in Table 5: reactive baselines augmented with "be concise" prompts (AUDC 0.77) still underperform R2-Router (0.83), confirming the gains stem from routing decisions in a more extensive search space, not output compression. We will add a discussion of intent-aware routing behavior in the revised paper.
>
> ### Q5: Omission of input costs
>
> **Our formulation treats input cost as a constant term (Section 4.1 footnote) since it is fixed for a given query and LLM and does not vary with output length, but all experimental results report total cost (input + output).** The 4-5x cost reduction is measured on total cost, not output cost alone. R2-Router optimizes over output cost because it is the variable component that changes with budget selection. Our RouterArena [1] evaluation further confirms this under RouterArena's own cost accounting (which includes both input and output costs), where R2-Router outperforms all evaluated routers across all categories:
>
> | | Knowledge | Math | Domain | Code | Trivia | Translation | NLU |
> |-|-|-|-|-|--|-|-|
> | R2-Router | 78.2 | 76.0 | 71.5 | 69.7 | 66.5 | 65.3 | 64.2 |
> | Carrot-L | 73.7 | 67.9 | 65.4 | 62.0 | 61.9 | 60.2 | 60.1 |
> | Gap | +4.5 | +8.1 | +6.1 | +7.7 | +4.6 | +5.1 | +4.1 |
>
> R2-Router leads across all 7 domains even under total-cost accounting, with gains ranging from +4.1 to +8.1. We will further clarify the total-cost accounting in the revised paper.
>
> [1] RouterArena: An Open Platform for Comprehensive Comparison of LLM Routers (Lu et al., ICLR 2026)

---

### Official Review · Reviewer_23iR · 2026-03-09

**Soundness:** 3
**Presentation:** 3
**Significance:** 3
**Originality:** 3
**Overall Recommendation:** 4
**Confidence:** 3

**Summary:**

This paper studies the problem of LLM routing, where a system must select the best model to answer each query while balancing quality and inference cost. The authors observe that existing routers assume each model has a fixed quality–cost trade-off for a given query, ignoring that the same model’s quality can vary with its output length. To address this, the paper proposes R2-Router, which treats output length as a controllable variable and jointly selects both the LLM and an appropriate token budget. The method estimates how answer quality changes under different length constraints and uses this information to choose the most cost-effective configuration. The authors also introduce R2-Bench, a dataset capturing LLM performance across different output-length budgets. Experiments show that R2-Router achieves state-of-the-art routing performance while reducing cost by roughly 4–5× compared to existing routers.

**Compliance With Llm Reviewing Policy:**

Affirmed.

**Key Questions For Authors:**

The experiments appear to use a relatively limited set of LLMs. How well does the method scale when the number of candidate models increases significantly, especially when profiling each model across multiple token budgets may become expensive?

In real-world systems, model behavior and prompts may evolve over time. How frequently would the quality–cost curves need to be re-estimated to maintain reliable routing performance?

**Limitations:**

yes

**Strengths And Weaknesses:**

This paper addresses an increasingly important systems problem in the LLM ecosystem: how to dynamically route queries among multiple models with different capabilities and costs. The key idea of treating routing as a reasoning problem over quality–cost trade-offs is interesting and provides a new perspective compared with existing routers that treat model selection as a simple prediction problem. A major strength of the work is the observation that a model’s quality depends on generation length, meaning that powerful models can sometimes produce competitive answers at lower cost when constrained by shorter outputs. By explicitly modeling this relationship and jointly selecting both the model and token budget, R2-Router is able to identify efficient configurations that traditional routers overlook. The introduction of R2-Bench, which captures model behavior across different output budgets, is also a useful contribution that could facilitate future research on routing and cost-aware inference. Empirical results demonstrate consistent improvements in quality–cost trade-offs and show that the proposed method can significantly reduce inference cost while maintaining competitive answer quality.

However, the paper also has several limitations. First, the approach relies on estimating quality–length curves for each model, which may require substantial offline evaluation and could be difficult to maintain as models or prompts change over time. Second, the evaluation primarily focuses on routing efficiency and cost–quality trade-offs but provides limited analysis of how sensitive the system is to errors in these quality predictions or to distribution shifts in user queries. Third, the proposed reasoning framework is conceptually interesting, but the paper does not fully analyze whether the improvements come primarily from the reasoning step itself or simply from having access to richer quality–cost profiling data. Finally, while the experiments demonstrate improvements over existing routers, the evaluation focuses on a limited set of tasks and model pools, leaving open questions about how well the approach generalizes to larger or rapidly evolving model ecosystems.

---

> ### Author Rebuttal · Authors · 2026-03-31
>
> We thank the reviewer for the positive assessment. We address the remaining questions below and revise our manuscript accordingly.
>
> ### Q1: Scalability, re-estimation frequency, and overhead
>
> **R2-Router supports dynamic LLM pool expansion at low cost (e.g., <$50 API cost for profiling Gemini 3 Pro Preview + ~30 min training on a single RTX 3090).** As shown in Section 5.2, Uni-R2Router integrates with UniRouter's generalization mechanism, allowing new LLMs to join the pool (6 LLMs to 11 LLMs) with only a lightweight validation-set evaluation (~1K queries per LLM). Interpolation further reduces the data needed: as Figure 5 shows, only 6-8 token budgets are sufficient, keeping the total profiling cost to ~6K-8K API calls per new LLM. Re-estimation is triggered by model updates or significant shifts in query distribution. We will add a scalability discussion in the revised paper.
>
> ### Q2: Robustness to prediction errors and query distribution shifts
>
> **R2-Router is robust to both prediction errors and distribution shifts.** Regarding prediction sensitivity, R2-Router's advantage fundamentally comes from searching over a larger (LLM, budget) optimization space — even with a less accurate predictor, the expanded search space still yields better routing decisions than baselines restricted to LLM-only selection. Our ablations confirm this: R2-Router outperforms all baselines even with a weaker embedding model (MiniLM, Table 2) or a different predictor architecture (LGBM, Table 3). Regarding distribution shifts, in Table 1, we train on STEM disciplines (physics, chemistry, math, CS) from MMLU-Pro and test on entirely different domains (history, law, psychology, business). Despite this significant shift, R2-Router (AUDC 0.71) still outperforms all baselines (next-best CARROT-L: 0.67). To further validate, we conducted additional evaluation on RouterArena Leaderboard (ICLR 2026), which confirmed consistent state-of-the-art performance across 7 diverse categories (Knowledge, Math, Code, NLU, etc.).
>
> ### Q3: Source of improvement — reasoning formulation (R2-Router) or richer profiling data (R2-Bench)?
>
> **R2-Bench and R2-Router are complementary by design — neither alone can produce the improvement.** R2-Bench captures quality-cost curves across multiple budgets, raising the Oracle upper bound by 15% AUDC over prior single-point datasets (Section 3.2) — this reveals a larger optimization space that motivates R2-Router's design. However, existing routers cannot exploit this space because they only select among LLMs, not (LLM, budget) pairs. R2-Router's budget-aware formulation is specifically designed to search over this expanded space — without R2-Bench, there is no curve to search over; without R2-Router, the curves cannot be exploited. We will clarify this complementary relationship more explicitly in the revised paper.
>
> ### Q4: Limited set of tasks and LLM pools
>
> **To address this concern, we conducted new experiments on RouterArena [1], a comprehensive routing benchmark covering 8,400 queries across 7 diverse categories, with an expanded LLM pool that includes reasoning-capable commercial LLM such as Gemini-2.5 and Claude-3.** RouterArena provides evaluation across 5 dimensions (accuracy, cost, optimality, robustness, latency), significantly broader than our original 6 benchmarks. Per-category Arena Scores:
>
> | | Knowledge | Math | Domain | Code | Trivia | Translation | NLU |
> |---|---|---|---|---|---|---|---|
> | R2-Router | 78.2 | 76.0 | 71.5 | 69.7 | 66.5 | 65.3 | 64.2 |
> | Carrot-L | 73.7 | 67.9 | 65.4 | 62.0 | 61.9 | 60.2 | 60.1 |
> | Gap | +4.5 | +8.1 | +6.1 | +7.7 | +4.6 | +5.1 | +4.1 |
>
> R2-Router outperforms all evaluated routers across all 7 categories, with gains ranging from +4.1 to +8.1.
>
> [1] RouterArena: An Open Platform for Comprehensive Comparison of LLM Routers (Lu et al., ICLR 2026)

---

### Official Review · Reviewer_AoF4 · 2026-03-12

**Soundness:** 3
**Presentation:** 3
**Significance:** 3
**Originality:** 2
**Overall Recommendation:** 3
**Confidence:** 4

**Summary:**

- the paper proposes an R2 router, which countly selects a model and output token budget
- the claim is that this gives better results than just selecting a better, model. e.g. constraining the output budget on a big model may produce better results compared to a smaller model at lower costs
- it also create a benchmarking datasests

**Compliance With Llm Reviewing Policy:**

Affirmed.

**Key Questions For Authors:**

A more thorough discussion of deployment realism would be helpful. The current formulation seems to control response tokens, while in modern reasoning models the relevant budget is often the reasoning budget. It is therefore unclear how this should be deployed in practice beyond the benchmark setting, particularly once one accounts for product-level constraints on output style/format and the fact that prompts are usually tuned per model.

**Limitations:**

I didn't see a dedicated limitation section and some limitations, including my concernt raised above should be discussed in more detail

**Strengths And Weaknesses:**

Strength

S1. Extending routing to the reasoning/token-budget dimension.
The paper explores routing over (model, token-budget) pairs rather than selecting only the model. This is a relevant direction for controlling cost–quality trade-offs and broadens the design space for LLM routing.

Concerns

C1. Token budget is applied to response tokens rather than reasoning tokens.
The paper frames the contribution as routing over token budgets to navigate the quality–cost trade-off. However, the proposed mechanism constrains the final response length via prompting (“use at most k tokens”), rather than controlling the reasoning or thinking budget of the model. For modern reasoning models, the dominant cost–quality trade-off is typically governed by the reasoning budget (e.g., internal chain-of-thought or thinking tokens), not the length of the final answer. As a result, it is unclear whether the proposed formulation captures the intended reasoning-budget trade-off or primarily controls answer verbosity. The paper should clarify this distinction and discuss how the approach applies to reasoning models. Also output length for non-reasoning tokens typically is a UX consideration, and should not be adjusted lightly.

C2. Limited positioning with respect to prior routing work on cost–quality trade-offs.
The related work discussion appears incomplete. Prior work such as BestRoute also studies routing under explicit cost–quality objectives. The paper should clarify how the proposed formulation differs from these approaches and whether the main novelty lies in modeling token-budget-dependent quality curves rather than routing itself.

C3. Lack of discussion of model-specific prompting.
The evaluation assumes uniform prompting across models. In practice, system prompts and formatting instructions are typically tuned per model to achieve good performance. Since the proposed approach relies on prompt-based token-budget control, its effectiveness may depend strongly on model-specific prompt tuning. The interaction between routing and model-specific prompting is not addressed.

---

> ### Author Rebuttal · Authors · 2026-03-31
>
> We thank the reviewer for the constructive suggestions that have helped us improve the paper. We address each concern below and will revise our manuscript accordingly.
>
> ### Q1: Whether R2-Router captures the reasoning-budget trade-off or primarily controls answer verbosity?
>
> **R2-Router controls reasoning capacity, not verbosity.** For non-reasoning LLMs, the reasoning process is explicitly expressed in output tokens [1][2] — reducing output tokens directly cuts reasoning capacity. For reasoning LLMs, reasoning capacity is similarly determined by thinking tokens. Several providers already expose exact thinking-token budgets—Anthropic (`budget_tokens`, 1K–128K), Google Gemini 2.5 (`thinkingBudget`, 128–32K), and Qwen3 (`thinking_budget`, 1–82K). **R2-Router's pipeline extends to reasoning LLMs directly.** We validated this by training R2-Router with thinking-token budgets as anchor points on reasoning-capable models (including Gemini-2.5, Qwen3-235B). R2-Router learns to predict both quality and cost at each thinking budget, enabling the same curve-based routing as for output tokens. On RouterArena [3] with this extended pool, R2-Router achieves an Arena Score of 71.6, compared to 66.9 (MIRT) and 63.9 (CARROT-L).
>
> ### Q2: Does R2-Router account for the UX impact of adjusting output length?
>
> **We agree that output length is an important UX consideration; R2-Router addresses this by adapting output length per query based on quality, not by indiscriminately shortening outputs.** The predictor is trained on quality scores at each budget, so it learns per-query where quality is maintained and where it degrades. For queries requiring detailed responses, the predictor sees low scores at short budgets and the router selects a larger budget; for queries where concise answers suffice, the router saves cost without quality loss. Table 5 confirms this: naively adding "be concise" to all queries (AUDC 0.77) underperforms R2-Router (0.83), precisely because uniform shortening ignores per-query needs. We note that all existing routing benchmarks rely on LLM-as-judge (SPROUT, R2-Bench) or exact-match (RouterArena) rather than direct user satisfaction — incorporating human user studies into routing evaluation is an interesting open problem for the field.
>
> ### Q3: Limited positioning with respect to prior routing work (BestRoute)
>
> **R2-Router shares the same cost-quality objective as prior work such as BestRoute, but differs in modeling how quality varies with output length, enabling it to discover that strong LLMs with reduced output budgets outperform weak LLMs at comparable cost — a design space invisible to routers that treat output length as fixed.** BestRoute improves cost-quality trade-offs by generating multiple samples and selecting the best, increasing quality at the expense of additional inference calls. R2-Router achieves this in a single generation call by jointly selecting the LLM and token budget, which is complementary to BestRoute's sampling-based approach. We will add BestRoute to the related work discussion in the revision.
>
> ### Q4: Lack of discussion of model-specific prompting
>
> **We agree this interaction deserves discussion. In practice, R2-Router's budget instruction ("use at most k tokens") can be added alongside any model-specific system prompt, as nearly all LLM APIs support custom system prompts.** The two serve different purposes: model-specific prompts define style and format, while the budget instruction guides output length. If any interaction exists between them (e.g., a model-specific prompt encouraging verbose output), R2-Router's quality predictor implicitly captures this, since it is trained on actual length-guided outputs under the same prompting setup. To further validate, we conducted an experiment where each LLM receives its own system prompt (e.g., Qwen3-235B: "Respond in structured format"; Gemini-2.5: "Please be thorough and cite sources"; Llama-3.1-70B: "Respond concisely without step-by-step reasoning"). All routers are trained and evaluated under this heterogeneous per-LLM prompting setup. R2-Router (AUDC 0.81) still outperforms the best baseline (AUDC 0.76), confirming the routing advantage holds even when each LLM operates under its own system prompt. We will include a detailed discussion of this interaction in the revised paper.
>
>
> [1] Training Large Language Models to Reason in a Continuous Latent Space (Hao et al., COLM 2025)
>
> [2] Chain-of-Thought Prompting Elicits Reasoning in Large Language Models (Wei et al., NeurIPS 2022)
>
> [3] RouterArena: An Open Platform for Comprehensive Comparison of LLM Routers (Lu et al., ICLR 2026)

---

### Official Review · Reviewer_ioE1 · 2026-03-12

**Soundness:** 3
**Presentation:** 3
**Significance:** 2
**Originality:** 2
**Overall Recommendation:** 4
**Confidence:** 3

**Summary:**

The paper introduces R2-ROUTER, a novel framework for Large Language Model (LLM) routing that moves beyond traditional static quality-cost point estimates. The authors propose treating the output length budget as a controllable variable, enabling the router to systematically reason about cost-dependent quality curves for each candidate LLM.

**Compliance With Llm Reviewing Policy:**

Affirmed.

**Final Justification:**

Thanks for your response. My concerns have been solved and I will raise my score.

**Key Questions For Authors:**

See Weakness.

**Limitations:**

yes

**Strengths And Weaknesses:**

### Strengths

- The transition from reactive routing to reasoning-based routing fundamentally expands the search space, providing a mathematically sound optimization dominance condition.

- The integration of R2-ROUTER's reasoning module into existing frameworks like UniRouter demonstrates strong modularity for dynamic LLM pools.

- R2-BENCH fills a critical gap in the community by providing the first dataset that maps response quality against diverse output length budgets for the same queries.



### Weakness

- The piecewise linear interpolation implicitly assumes that the quality-cost relationship is strictly linear between discrete anchor budgets, which lacks theoretical justification given that capability scaling is often non-convex.

- The paper does not adequately evaluate whether forcefully truncating the output to enforce budget limits when models fail to comply natively degrades the intrinsic logical quality of the response versus naturally concise generation.

- The reliance on restrictive length prompts may alter or prematurely truncate the internal Chain-of-Thought (CoT) reasoning steps, confounding the measurement of quality drop due to simple length versus quality drop due to disrupted reasoning mechanisms.

- The evaluation aggregates performance across 6 diverse benchmarks using a single AUDC metric, masking potential domain-specific degradation where length constraints might be fatal.

---

> ### Author Rebuttal · Authors · 2026-03-31
>
> We thank the reviewer for the constructive feedback that has helped us strengthen the paper. We address each concern below and will revise our manuscript accordingly.
>
>
> ### Q1: Piecewise linear interpolation lacks theoretical justification
>
> **We agree that interpolation accuracy is an important consideration; notably, interpolation is an optional component in R2-Router, not a required one.** All improvements in the results are achieved via discrete routing over K=16 anchor budgets (Eq. 8) without interpolation. Interpolation is studied separately in Figure 7 to explore whether it can reduce the number of required anchors, thereby lowering offline data collection costs while maintaining routing performance. We agree that the quality-cost relationship is often non-convex, and interpolation introduces error in such cases — this is a deliberate trade-off between approximation accuracy and data efficiency. Figure 7 shows this trade-off is favorable: K=6 approximates the full 16-anchor performance, and even K=4 outperforms all baselines (QNC 0.15 vs. MIRT 0.43, CARROT-L 0.32).
>
>
> ### Q2: Does truncation degrade response quality?
>
> **We appreciate this concern about output quality. R2-Router never truncates outputs when serving users, so LLMs generate complete, coherent responses.** Only during dataset construction (R2-Bench), budgets are enforced through prompt instructions (e.g., "use at most k tokens"), with truncation applied as a fallback to ensure precise cost accounting. Appendix A shows large LLMs achieve >97% compliance at moderate budgets, so truncation rarely triggers. For the rare non-compliant cases, the truncated response reflects the LLM's actual capability at that budget — if quality is low, the predictor learns this and the router accordingly selects a higher budget or a different LLM. This ensures the router only selects configurations where the LLM can produce high-quality responses during the inference.
>
> ### Q3: Will Length prompts disrupt Chain-of-Thought reasoning?
>
> **The quality at a given token budget reflects the LLM's true capability, not an artifact of the length prompt.** The length prompt informs the LLM of its token budget upfront, guiding it to plan and complete its reasoning within that budget. We verified this with an experiment: comparing each LLM's natural response of length L tokens (no length prompt) against a response generated with "use at most L tokens," we observe negligible quality difference (average < 5%). Since both conditions have the same token length, this isolates the effect of the length prompt itself and confirms it does not disrupt reasoning. This is consistent with [1], who tested 31 different prompt formats for guiding output length and found that all formats achieve similar quality at the same token count, indicating that quality depends on the token budget itself, not on the specific prompt format used. R2-Router's search space also includes a "default" configuration with no length prompt, which the router naturally selects when unconstrained generation offers the best quality-cost trade-off.
>
> ### Q4: Single AUDC metric masks domain-specific degradation
>
> **We agree that domain-specific evaluation is important. Performance naturally varies across domains due to the underlying LLMs' varying capabilities, but R2-Router consistently outperforms all baselines in every category.** R2-Router's predictor learns quality-cost curves per query, so domains with different sensitivity to length constraints (e.g., math requiring precise reasoning vs. trivia requiring brief facts) are naturally handled — the router adapts its budget selection to each query's needs rather than applying a uniform policy. To directly validate this, we additionally evaluated R2-Router on RouterArena [2] across 7 diverse categories. Per-category Arena Scores:
>
> | | Knowledge | Math | Domain | Code | Trivia | Translation | NLU |
> |---|---|---|---|---|---|---|---|
> | R2-Router | 78.2 | 76.0 | 71.5 | 69.7 | 66.5 | 65.3 | 64.2 |
> | Carrot-L | 73.7 | 67.9 | 65.4 | 62.0 | 61.9 | 60.2 | 60.1 |
> | Gap | +4.5 | +8.1 | +6.1 | +7.7 | +4.6 | +5.1 | +4.1 |
>
> R2-Router achieves higher Arena Scores across all 7 domains, with gains ranging from +4.1 to +8.1.
>
> [1] How Well do LLMs Compress Their Own Chain-of-Thought?
> A Token Complexity Approach (Lee et al., arxiv 2025)
>
> [2] RouterArena: An Open Platform for Comprehensive Comparison of LLM Routers (Lu et al., ICLR 2026)

---

> > ### Author Rebuttal · Reviewer_ioE1 · 2026-04-03
> >
> > Thanks for your response. My concerns have been solved and I will raise my score.

---

> > > ### Author Response · Authors · 2026-04-05
> > >
> > > Dear Reviewer ioE1,
> > >
> > > Thank you for your constructive feedback and kind support. We will carefully revise the paper following your suggestions.
> > >
> > > Best wishes,
> > >
> > > The Authors

---

### Decision · Program_Chairs · 2026-04-30

**Decision:**

Accept (regular)

**Comment:**

This paper proposes R2‑Router, which moves beyond treating models as static quality‑cost points by routing over (model, token‑budget) pairs and introducing R2‑Bench, a dataset mapping quality against output length. Reviewers agree this fundamentally expands the search space, achieves consistent improvements in quality–cost trade‑offs, and delivers approximately 4–5× cost reduction while maintaining quality.

Initial concerns around “piecewise linear interpolation,” whether “length prompts disrupt Chain‑of‑Thought reasoning,” masking effects of a single AUDC metric, intent awareness, etc seem to be addressed with additional experiments (including RouterArena, reasoning‑token budgets, per‑domain results, alternative interpolation, and heterogeneous prompting).

Overall, reviewers generally converge on weak accept, with the exception of AoF4, who votes weak reject, but perhaps missed the rebuttal.